# Effects of Distracting Behaviors on Driving Workload and Driving Performance in a City Scenario

**DOI:** 10.3390/ijerph192215191

**Published:** 2022-11-17

**Authors:** Shuang Luo, Xinxin Yi, Yiming Shao, Jin Xu

**Affiliations:** 1College of Traffic and Transportation, Chongqing Jiaotong University, Chongqing 400074, China; 2Chongqing Chang’an Automobile Co., Ltd., Chongqing 400023, China

**Keywords:** distracted driving, city route, real driving condition, heart rate variability, driving performance

## Abstract

Distractors faced by drivers grow continuously, and concentration on driving becomes increasingly difficult, which has detrimental influences on road traffic safety. The present study aims to investigate changes in driving workload and driving performance caused by distracting tasks. The recruited subjects were requested to drive along a city route in a real vehicle and perform three secondary tasks sequentially. Electrocardiography and driving performance were measured. Heart rate variability (HRV) was adopted to quantitatively analyze the driving workload. Findings show that: (i) increments are noticed in the root mean square differences of successive heartbeat intervals (RMSSD), the standard deviation of normal-to-normal peak (SDNN), the heart rate growth rate (HRGR), and the ratio of low-frequency to high-frequency powers (LF/HF) compared to undistracted driving; (ii) the hands-free phone conversation task has the most negative impacts on driving workload; (iii) vehicle speed reduces due to secondary tasks while changes in longitudinal acceleration exhibit inconsistency; (iv) the experienced drivers markedly decelerate during hands-free phone conversation, and HRGR shows significant differences in both driving experience and gender under distracted driving conditions; (v) correlations exist between HRV and driving performance, and LF/HF correlates positively with SDNN/RMSSD in the hands-free phone conversation and chatting conditions while driving.

## 1. Introduction

Road traffic accidents are the third leading cause of unnatural deaths, following psychiatric illnesses and cardiovascular diseases. About 1.3 million lives are ended as a result of road traffic accidents every year, and a range of 20–50 million people suffer injuries and disability [1]. It is reported that the majority of road traffic accidents are contributed to drivers, and distracted driving is the most important contributing factor [2].

Distracted driving refers to performing secondary activities including calling, texting, voice messaging, operating navigation, eating, drinking, etc. while driving [3]. Distracting behaviors attract driver’s attention from the primary driving task to the secondary tasks, which is risky and accident-prone. Therefore, distracted driving is a concern worldwide. Analysis data corresponding to distraction are mainly derived from naturalistic driving data [4,5,6], simulator-based, and real driving experiments.

Driving simulator experiments provide safe and controlled scenarios and allow a variety of risky secondary tasks. Mobile phone use is a common distracting behavior while driving, including making a phone call, voice messaging, texting, and browsing social media. The usage of hands-free and hands-held phone calls seem to have a similarly negative influence on driving performance [7], as opposed to chatting with an in-vehicle person [8]. The above-mentioned statement was confirmed by Choudhary et al. [9], in which all drivers thought it not at all risky to converse with the passengers in the vehicle. Voice messaging is more distraction-prone than a mobile phone call, as drivers are not aware of risks generated by voice messaging while driving [10]. Voice messaging significantly decreases the standard deviation of lateral position between two vehicles and increases collision rates compared to undistracted driving [11]. The attention location during browsing Facebook shows a similarity to that while texting, but browsing Facebook is not as detrimental as texting when driving [12]. A similar conclusion was drawn by Choudhary et al. [9], in which more than 70% of 90 drivers reported texting was an extremely risky behavior while driving.

Real driving tests are not as safe as the simulator-based tests. The secondary tasks in the real driving conditions are less distracting and detrimental. Looking at the particular target from the windscreen, rearview mirrors, and windows when driving is related to the primary driving task, which results in visual distraction and lane departure [13,14]. The reaction time of drivers is postponed while conversing through a mobile phone on a motorway [15], which shows an agreement with the result obtained in a field test [16]. Reading text in a real driving scenario leads to the vehicle departing the road, and these risky experiments were carried out in a track environment [17]. In a city route, drivers were asked to conduct five secondary tasks including radio operating, GPS operating and following, mobile phone calling, picture describing, and conversing with an in-vehicle passenger to investigate drivers’ visual and cognitive distractions and the changes of driving behaviors [18,19].

Driving performance and psychophysiological responses are impacted while driving under a distracted condition. Driving performance indicators are easily measured and widely adopted to investigate distracted driving. Vehicle speed reduces when performing secondary tasks [20,21,22,23]. Ebnali et al. [24] reported a higher speed variability was caused, while Reimer et al. [20] obtained a lower standard deviation of speed. Kim and Yang [25] showed vehicle lateral movement was increased, while other research by Bowden et al. [21] reported the lane maintenance capacity was slightly enhanced. The bilateral prefrontal and parietal cortical activity significantly increased on account of smartphone distraction [26]. However, Wester et al. [27] highlighted the processing of irrelevant and distracting secondary tasks by the cerebral cortex decreases while driving. The electroencephalography power and the response time of secondary tasks both change noticeably under different stimulus [28]. Moreover, a driver’s reaction time to the primary tasks, e.g., braking, turning on lights, and traffic light response, is increased as well due to the distracting behaviors [12,15,16,29,30,31].

In summary, most of the existing distracted driving data have derived from simulator-based driving experiments [7,8,9,10,11,12,20,21,22,23,25,26,27,28]. Although similarity to a real driving condition exists, there are still great differences [32]. Furthermore, driving simulators are not the best apparatus for measuring driving behaviors [33], especially for Electrocardiography (ECG) signals. Real driving experiments have been conducted on motorways [13,14,15], controlled fields, and track environment [16,17], but distracted driving tests in real conditions under complex urban scenarios are relatively few [18,19]. In addition, the majority of research concerned the differences in driving performance caused by distracting tasks, while driving workload that could be quantitatively analyzed by heart rate variability (HRV) was seldom involved.

This study is carried out to analyze the differences in HRV and driving performance between distracted and undistracted driving conditions. The distracting tasks were performed while driving along a city road in a real vehicle in sequential order: hands-free phone conversation, slogan reading, and chatting with an in-vehicle passenger.

This paper is organized as follows: in Section 2, the distracting tasks while driving are designed, and driving experiments are conducted; the main findings are shown in Section 3; the discussion is shown in Section 4; and conclusions are drawn in Section 5.

## 2. Materials and Methods

### 2.1. Subjects

There were 22 subjects with an average age of 24 years (range 20–32 years) and driving experience of 3.59 years (range 1–8 years), including 6 female and 16 male drivers, who participated in the real driving tests. All the subjects were in good physical condition, with normal uncorrected or spectacle-corrected visual acuity. Each subject fully understood the content and risk of the driving tests and signed the informed consent.

However, because of apparatus faults and failure to complete the distracting secondary tasks, 16 groups of effective experimental data were finally obtained, corresponding to 5 females and 11 males. Fundamental information of the 16 subjects is presented in Table 1.

### 2.2. Apparatus

A Honda SUV equipped with various sensors, as shown in Figure 1, was used in the driving experiment.

A Polyphysiograph (PhysioLAB) was employed to measure ECG signals of the drivers. The sampling rate of the PhysioLAB was chosen as 500 Hz in this study. Vehicle speed and acceleration were measured by the Speedbox, which was composed of a data acquisition unit and dual GPS (Global Positioning System) mounted on the vehicle’s roof. The sampling rate was 20 Hz. Vehicle speed can also be obtained from the Mobileye630. Data redundancy and security are guaranteed by a combination of the Mobileye630 and the Speedbox. Two tachographs were mounted on the front windscreen. The one was employed to record the traffic environment and vehicle status outside, so as to understand the cause of abnormal experimental data and reject the unexpected values. The other one was used to record the driver’s distracting behaviors for the purpose of available data extraction.

### 2.3. Distracting Task

The design of distracting tasks should fully consider legislation allowance and driving risk. A hand-held phone conversation is prohibited by vehicular traffic laws in China. Texting, voice messaging, and navigation are regarded as extremely dangerous tasks and accident-prone during road tests in real vehicle scenarios. Therefore, the secondary tasks in this study are designed as follows.

Hands-free phone conversation (Phone): the subjects were requested to answer the phone after hearing the incoming call by pressing the button on the central screen and then converse continuously with the tester, who was not in the vehicle, for 1–2 min through the Bluetooth system on board. The subjects could refuse to answer the phone or terminate the conversation if risky.

Slogan reading (Slogan): the subjects were asked to look for the designated slogan with 10 Chinese characters and describe what was read to the tester. The slogan was located at the right side of the test road section, about 300 m behind a signal intersection. The slogan was 5 m in length, with a font size of 40 cm. The subjects were reminded after passing the intersection that there was a slogan in front. The subjects could reject to search for the slogan if it felt unsafe to do so.

Chatting with passengers (Chatting): the subjects chatted freely with the in-vehicle tester about work, daily life, and personal problems for approximately 1 min. The chat was initiated by the tester at a specified position.

### 2.4. Test Route

A city road section with six lanes in two directions and center median greenbelts located in the Nan’an District of Chongqing China was chosen as the test road, as shown in Figure 2. The test vehicle was determined to start from the Huilong Road South bus stop and drive south along main roadway, then make a U-turn at the intersection of Banan Avenue to head back to the Huilong Road South bus stop. The total mileage of the test route was approximate 7.4 km. The traffic volume was moderate, namely 3600–4500 veh/h, for the experimental duration. The test road section was spacious enough and there were no shelters above, which was a benefit for the acquisition of operational data experimentally.

### 2.5. Test Process

Each subject was asked to drive twice along the test route to compare the differences between normal conditions and distracting tasks in driving performance and HRV characteristics. In the first lap, the subjects did not conduct any distracting tasks. However, in the second lap, the subjects performed the distracting tasks in the order of “hands-free phone conversation, slogan reading and chatting with the tester in the vehicle”. The vehicle speed, longitudinal acceleration, and ECG of the drivers were measured in the tests. All subjects performed the same secondary task on the same road section, so that the built environment, road condition, traffic signals, and speed limitation, etc., could be basically kept fixed in the two laps of the same subject [18,19]. Therefore, the differences in driving performance and HRV characteristics were mostly caused by the distracting driving behaviors.

All the tests were conducted at the off-peak time in the morning and afternoon, to avoid the impact of traffic congestion on the experimental results. In addition, the weather and light conditions were fine, which guaranteed that the driving work was not hindered.

### 2.6. Data Processing

The available data refer to the experimental measurements during normal conditions and distracting tasks on the road sections when performing the designated distracting task. The experimental results, thus, must be separated and extracted.

The duration of each distracting task could be determined via the video from the tachographs. The data acquisition time of the Speedbox, Mobileye630, PhysioLAB, and tachographs needed to be unified to ensure time synchronization of multi-source data. For this purpose, the beginnings of data acquisition of the Speedbox, Mobileye630, and PhysioLAB were calculated according to the end moment of the command of “Driving Test Begins” given by the tester. Time and speed data were both employed to enhance the accuracy.

The data segmentation and extrication method of one subject is shown in Figure 3. The tachographs were operated and started recording first. The sections D1, D2, and D3 individually represented the retardation time in the starting measurement of the PhysioLAB, the Mobileye630, and the Speedbox relative to the tachographs. The hands-free phone task began after the vehicle started driving for a period of time that was indicated by section D4. Section D5 denoted the time interval between the hands-free phone conversation and slogan reading task, while D6 indicated the time interval between slogan reading and chatting task. Sections D7, D8, and D9 represented the durations of hands-free phone conversation, slogan reading, and chatting, respectively. P-time, M-time, and S-time indicated the task durations recorded by the PhysioLAB, Mobileye630, and Speedbox, respectively. Moreover, *v*_m1_ and *v*_m2_ individually denoted vehicle speed measured by Mobileye630 at the start and end time of the task durations, while *v*_s1_ and *v*_s2_ were measured by Speedbox. The starting and ending station of each secondary task were recorded by the tachograph, according to which the experimental results in normal conditions were obtained for comparison.

It should be mentioned that reading the slogan on the roadside only cost several seconds, which resulted in a difficulty in analyzing HRV characteristics. Therefore, the duration of slogan reading started from the moment when the subject was reminded a slogan existed in front of them, and also involved the distraction recovery for a 10 s period [21]. In addition, a Savitzky–Golay filter was used to eliminate any noise of the experimental data. Savitzky–Golay is a filter technique based on least squares fitting, which reconstructs waveforms with less computational complexity and marginal distortion [34,35].

## 3. Results and Discussion

HRV characteristics, including the heart rate growth rate (HRGR), the standard deviation of normal-to-normal peak (SDNN), the root–mean–square differences of successive heartbeat intervals (RMSSD), the ratio of low-frequency to high-frequency powers (LF/HF) [36,37], and driving performances involving vehicle speed and longitudinal acceleration, were adopted as trial indicators. It was found that the vehicle speed corresponded to normal distribution by Shapiro–Wilk normality test. Therefore, the independent-sample *t* test was used for this indicator, while Mann–Whitney U test was used for the others.

### 3.1. HRV Characteristics

#### 3.1.1. SDNN

SDNN indicates that a variation in heart rate is a typical and commonly used indictor for analyzing HRV in time domain. The boxplot of SDNN under different driving conditions is presented in Figure 4. The upper quartile, mean, and median of SDNN increased while distracted driving. At the mean level, SDNN increased by 61.17% in the hands-free phone conversation condition while driving, 42.59% under the slogan reading task, and 8.92% while chatting, among which the SDNN of drivers during hands-free phone conversation was affected the most. The fluctuation ranges of SDNN while chatting and slogan reading were roughly the same, but less than that during hands-free phone conversation. Statistical analysis showed there was a significant difference in SDNN between the normal condition and hands-free phone conversation (*p* < 0.05).

#### 3.1.2. RMSSD

RMSSD is a significant index for atrial fibrillation. Figure 5 shows the RMSSD during distracted and normal conditions. The average RMSSD increased by 85.07%, 21.14%, and 4.93% during hands-free phone conversation, slogan reading, and chatting. There was no significant difference in RMSSD between the normal conditions and any distracted ones (*p* > 0.05). Although, RMSSD in hands-free phone conversation was slightly larger than that of the normal driving condition, with a significance level of 0.1.

#### 3.1.3. HRGR

HRGR reports growths in heart rate during driving process. The boxplot of HRGR is presented in Figure 6. The HRGR of drivers during secondary tasks was separately greater than that of normal conditions, indicating HRGR that increased with a growth of psychological load. In comparison to normal driving conditions, the mean HRGR in the distracted situation respectively increased by 110.56%, 162.12%, and 31.31%, among which HRGR increased the most during slogan reading, followed by hands-free phone conversation. The box size showed the greatest difference in individual attributes during hands-free phone conversation. There were significant differences in HRGR between all the distracting tasks and the corresponding normal conditions, respectively (*p* < 0.05).

#### 3.1.4. LF/HF

LF/HF is obtained by a time-frequency transformation of HRV. It is positively related to the driver’s mental workload, which indicates LF/HF increases with an increase in driver’s mental workload or tension degree.

In comparison to the normal conditions, the lower quartile, mean, median, and upper quartile of LF/HF all increased during distracted driving, as shown in Figure 7. The mean LF/HF of the drivers individually increased by 16.1% and 23.8% in the slogan reading and chatting tasks, while they increased by about 146.4% during hands-free phone conversation. At the level of box size, LF/HF fluctuated with the widest range in hands-free phone conversation compared with the other two distracted conditions. Statistical analysis showed significant increases in LF/HF during slogan reading and hands-free phone conversation in comparison to the normal conditions (*p* < 0.05), which indicated the two distracting behaviors were easily caused drivers’ tension while driving.

### 3.2. Driving Performance

Driving performance can reflect the differences in driving state between normal conditions and distracting tasks. In this study, vehicle speed and longitudinal acceleration were employed, which respectively represented the average value of all the instantaneous speed and longitudinal acceleration, during distracted driving and normal conditions.

#### 3.2.1. Speed

The boxplot of vehicle speed under the distracted conditions coupled with the corresponding normal ones is shown in Figure 8. The upper quartile, mean, median, and lower quartile of vehicle speed decreased when the drivers performed the distracting behaviors. The mean speeds were, respectively, 47.4 km/h, 46.4 km/h, and 48.9 km/h during hands-free phone conversation, slogan reading, and chatting, reduced by 9.2%, 7.2%, and 13.3% in comparison to the normal driving conditions. This indicated chatting had the greatest, while slogan reading had the least, impact on vehicle speed. Statistical analysis showed a significant difference between chatting and the normal condition (*p* < 0.05).

#### 3.2.2. Longitudinal Acceleration

The mean longitudinal accelerations were −0.1 m/s^2^ and 0.02 m/s^2^, individually, in the conditions of hands-free phone conversation and chatting, as shown in Figure 9; these decreased by 13.04% and 28.57% compared to the undistracted situations. The mean longitudinal acceleration was −0.51 m/s^2^ during slogan reading when driving, which increased by 23.53% relative to the normal driving condition. There was a significant difference in longitudinal acceleration between the chatting task and the normal condition while driving (*p* < 0.05).

### 3.3. Individual Attributes

The subjects were grouped according to gender and driving experience. The subjects with driving experience over four years were grouped into experienced drivers, and the other subject were novices. Statistical analyses were carried out in the distracted conditions with a significance level of 0.05. It was found that the longitudinal acceleration under hands-free phone conversation while driving showed a significant difference in the driving experience; HRGR under all distracted driving conditions exhibited significant differences in both gender and driving experience, except chatting in gender.

#### 3.3.1. Driving Experience

The longitudinal acceleration of novices was markedly greater than that of the experienced drivers (*p* = 0.002), as shown in Figure 10. This was mainly because some novice subjects were accelerating during driving with a hands-free phone conversation, and the longitudinal acceleration was positive, while the experienced subjects were decelerating and the longitudinal acceleration was negative.

The average rank of HRGR of the novices was 733 during hands-free phone conversation, which was 295 greater than that of the experienced subjects, as shown in Figure 11, and a significantly higher HRGR of the novice drivers was noticed (*p* = 0.009), which exhibited a similarity to the comparison obtained while driving with slogan reading was performed (*p* = 0.000). It indicated that the heart rate of the experienced drivers remained smoother than in the novices under the hands-free phone conversation and slogan reading conditions, which meant the novices were more burdened and felt more nervous during these secondary tasks.

However, HRGR of the novices was significantly lower than that of the experienced subjects under the chatting condition (*p* = 0.000). The HRGR of the novices mainly ranged from 8% to 24%, while the experienced subjects ranged from 16% to 32%, which indicated the experienced drivers felt more nervous during chatting. 

#### 3.3.2. Gender

During the hands-free phone conversation, the HRGR of male drivers was mainly distributed in 15–25% while, for female drivers, it was in 20–40%, as shown in Figure 12. During the slogan reading task, HRGR of the male and female drivers, respectively, ranged from 20% to 30% and 25% to 40%. The female drivers’ HRGR was generally higher than the male drivers’, and a significant difference existed between the male and the female drivers under the two distracted driving conditions (*p* = 0.000, *p* = 0.000), which indicated that the female drivers’ driving workload was higher and the females felt more tense.

### 3.4. Correlation Analysis

The Spearman correlation coefficient with the significance level of 0.05 was employed to analyze the relationship between driving performances and HRV characteristics during tasks. Results under the distracted driving conditions are shown in Table 2. No correlation was found between the driving performance indicators and the HRV indicators during hands-free phone conversation. A negative correlation existed between vehicle speed and RMSSD and between longitudinal acceleration and SDNN, and a positive relationship existed between longitudinal acceleration and LF/HF in the slogan reading condition while driving. Vehicle speed correlated positively with HRGR during the chatting task. There was a significantly positive correlation between RMSSD and SDNN, while negative correlation existed between RMSSD and LF/HF and between SDNN and LF/HF under all distracted driving conditions.

SDNN/RMSSD was calculated and correlation analysis between SDNN/RMSSD and LF/HF was carried out during tasks. The correlation coefficients were 0.410 (*p* = 0.015), 0.117 (*p* = 0.264), and 0.333 (*p* = 0.036) during hands-free phone conversation, slogan reading, and chatting, respectively, indicating a positive relationship between SDNN/RMSSD and LF/HF during the hands-free phone conversation and chatting tasks while driving. It seemed that the frequency domain indicator LF/HF could be roughly surrogated by the time domain characteristic SDNN/RMSSD for analyzing HRV of drivers during the two distracting tasks.

## 4. Discussion

In comparison to normal driving, SDNN, RMSSD, HRGR, and LF/HF increase during distracting tasks. Significant differences are found in SDNN during hands-free phone conversation, in HRGR during all secondary tasks, and in LF/HF during hands-free phone conversation and slogan reading. This indicates driver’s mental workload and tension degree are increased under distracted driving conditions and are the most impacted by hands-free phone conversation. The complexity of drivers’ brain function increases while processing dual tasks (driving and distracting behaviors) [26], which contributes to higher driving workload. The result showed a similarity to the previous studies [36,38], in which the authors highlighted that drivers’ physiological indictors grew with the processing of information in the traffic environment.

It could be noticed from the box size of the HRV characteristics that the hands-free phone conversation task was the most influenced by individual attributes. One possible explanation is that an unexpected incoming call heavily fluctuated the heartbeat of the driver who was concentrating on driving. Another possibility is that hands-free phone conversation is a combination of cognitive and manual (answer the phone) distractions. Driving risks were going to be greatest while drivers reach for in-vehicle objects [39].

The average speed decreased during tasks, which showed an agreement with previous studies. The vehicle driving speed significantly reduced when the drivers listened to the news and made an observation [24] or were engaged in a cell phone task [20] or navigation [23]. The drivers possibly noticed that the distracting tasks contributed to risky driving; they thus compensated driving performance by reducing the travelling speed. In addition, the lower speed ensured sufficient time left for the drivers to see the slogan clearly. The drivers initiated compensatory measures during distracted driving, resulting in a reduction in vehicle speed. The drivers compensated more while suffering greater driving risk on account of the responsibility for being safe [40]. Therefore, the largest speed reduction was exhibited during chatting, which indicated the greatest driving risk in this distracting task.

### 4.1. Practical Applications of Studies on Distracted Driving

Vehicle speed and acceleration are the most readily monitorable indicators that could be used to evaluate the distraction level and identify the distracted driving behaviors [41,42]. An increase in the driving workload is disclosed by the increase in heart rate while performing a distracting task [43]. However, the heart rate is affected by individual attributes. Individual differences could be avoided and the variation ranges of heart rate are presented more clearly by HRGR compared to the heart rate [20,36]. The driving performance indicators and HRV indictors could be adopted in the distraction detection model. Driving distraction detection and warning help to reduce traffic accidents and enhance driving safety. Furthermore, the study on distracted driving is beneficial for the deepened understanding of the detriment and risk caused by distractions. Therefore, findings in this study can provide a theoretical foundation for enhancing drivers’ safety awareness and improving risky driving behaviors and for the distraction warning system of an autonomous vehicle.

### 4.2. Limitation and Future Work

The experienced drivers were found to be more nervous in the chatting task from the HRGR. This seemed to be opposite to the life experience. However, the authors could not analyze the causes due to the limitation of knowledge and experiences. There might be insufficient samples.

Young subjects under 32 years old were recruited to participate in the tests. A large proportion of accidents were caused by the young drivers on account of insufficient driving experience and distraction-prone tendencies [44,45,46]. The young drivers displayed a greater deterioration in driving performance [40]. The differences in HRV and driving performance between young and old drivers would be an interest area of future research.

The hands-free phone conversation and chatting tasks were unexpected events, but the subjects knew what would occur before slogan reading during driving. This study did not present the differences in HRV and driving performance between secondary tasks, as these differences may be caused by the distinctions in tasks themselves, built environment, road conditions, etc., instead of whether the drivers know what would happen. The authors are planning to access the differences in experimental results between the expected and unexpected events, especially in the first few seconds, in future work.

## 5. Conclusions

The present study investigated distracted driving behaviors in real driving conditions on a city road. The results show that: (1) performing a secondary task while driving affects driving workload and driving performance, but not all the impacts are significant; (2) hands-free phone conversation have the greatest impact on HRV indictors while chatting on driving performance; (3) generally, the novices feel more burdened and nervous, and the female drivers are more tense during distracted driving; (4) correlations are found between HRV and driving performance in certain distracted driving conditions.

Distracted driving occupies visual, cognitive, and manual resources that safe driving requires. The mobility policies could propose that conversation with car drivers is avoidable either via the mobile phone or in-person. The drivers should be trained to realize the weaknesses and detrimental habits of driving and adopt compensatory measures to support safety of the vehicles while distracted. The experimental method including the design of distracting task, test process, and data processing is general and transferable to other situations, such as senior drivers, motorway, and rural scenarios. However, the experimental results in quantification may differ from those presented in this study due to different test scenarios and subjects. Further study will be carried out to comprehensively enhance the conclusions of this paper.

## Figures and Tables

**Figure 1 ijerph-19-15191-f001:**
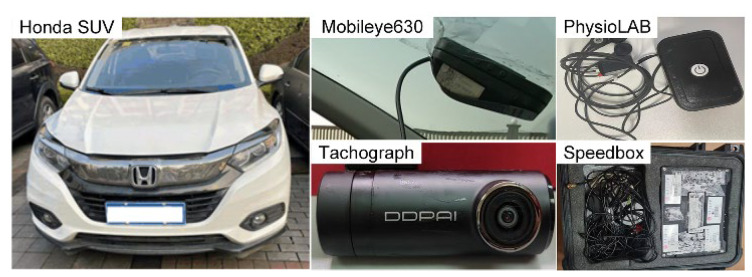
Experimental apparatus.

**Figure 2 ijerph-19-15191-f002:**
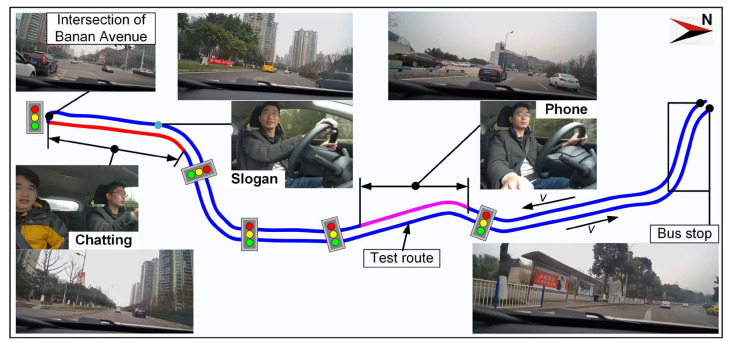
Test route and distracting task.

**Figure 3 ijerph-19-15191-f003:**
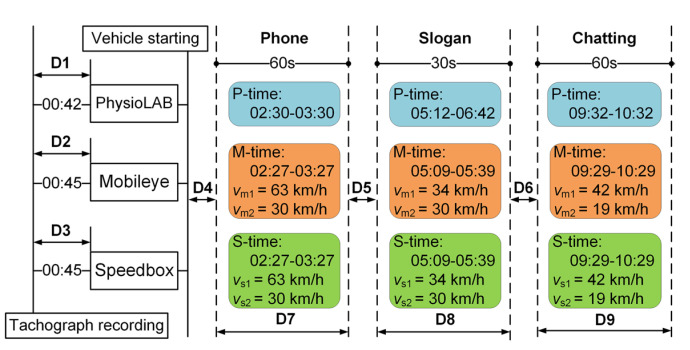
Extraction of available data.

**Figure 4 ijerph-19-15191-f004:**
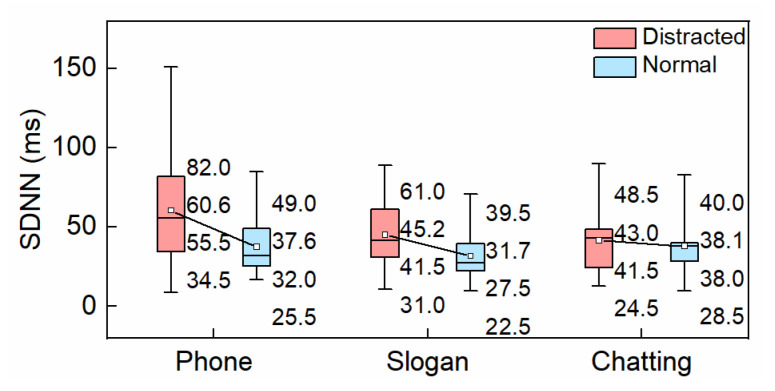
SDNN.

**Figure 5 ijerph-19-15191-f005:**
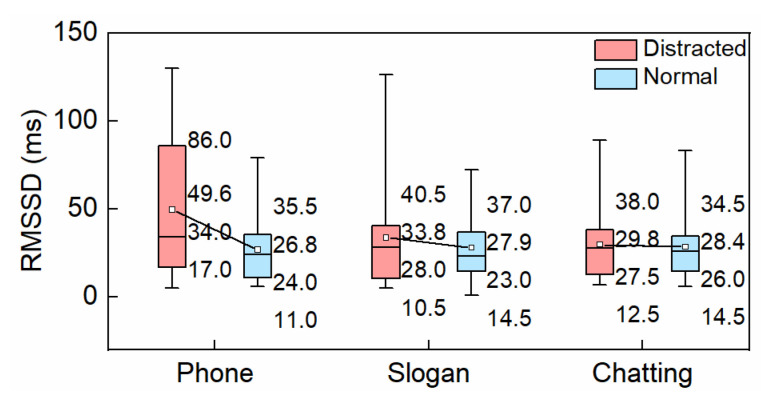
RMSSD.

**Figure 6 ijerph-19-15191-f006:**
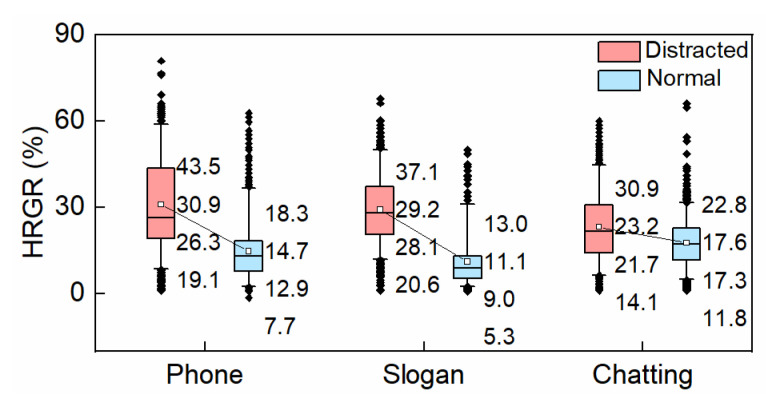
HRGR.

**Figure 7 ijerph-19-15191-f007:**
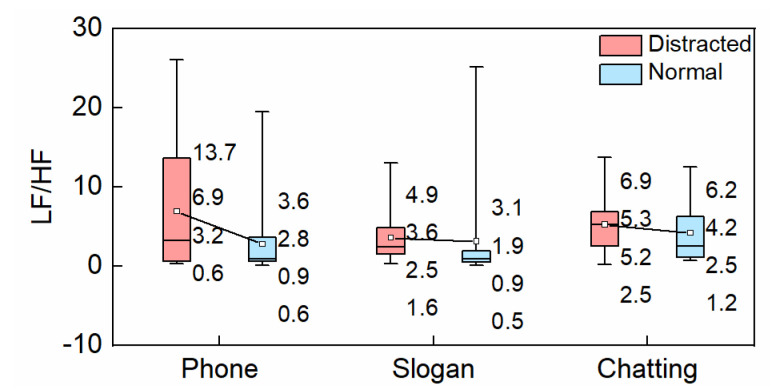
LF/HF.

**Figure 8 ijerph-19-15191-f008:**
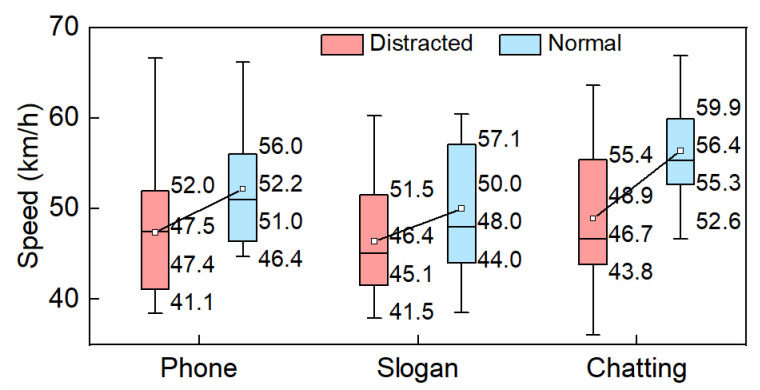
Vehicle speed.

**Figure 9 ijerph-19-15191-f009:**
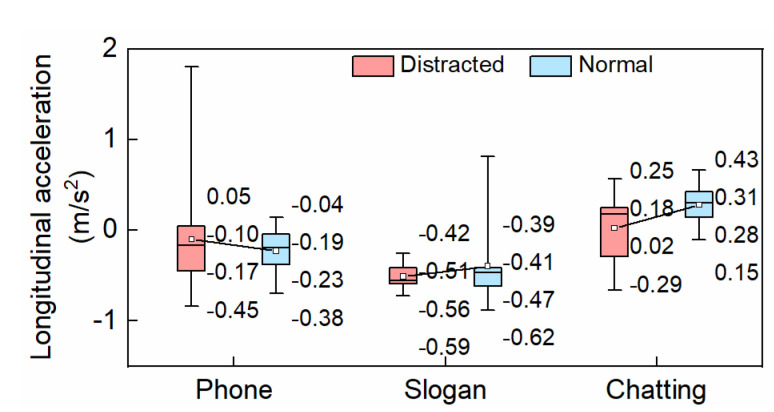
Longitudinal acceleration.

**Figure 10 ijerph-19-15191-f010:**
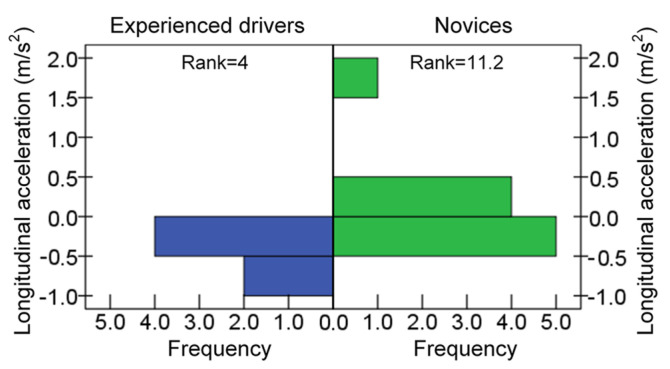
Difference in longitudinal acceleration.

**Figure 11 ijerph-19-15191-f011:**
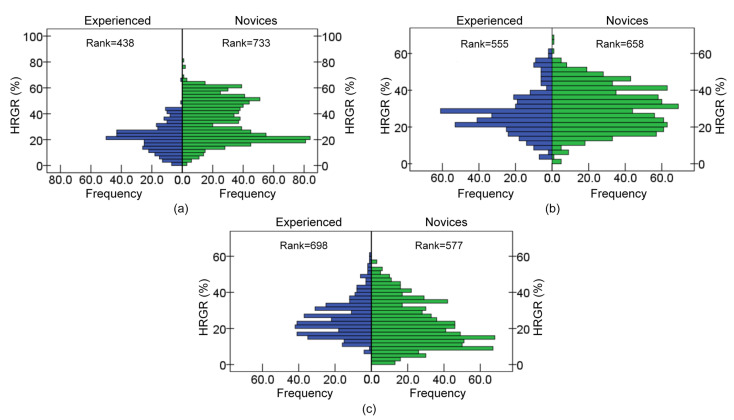
HRGR difference in driving experience during tasks: (**a**) Hands-free phone conversation, (**b**) slogan reading, and (**c**) chatting.

**Figure 12 ijerph-19-15191-f012:**
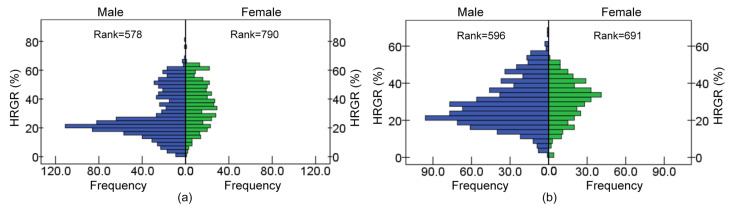
HRGR difference in gender during tasks: (**a**) Hands-free phone conversation; (**b**) slogan reading.

**Table 1 ijerph-19-15191-t001:** Fundamental information of subjects.

Subject	Gender	Driving Experience (Year)	Subject	Gender	Driving Experience (Year)
1	Male	6	9	Female	2
2	Male	7	10	Male	8
3	Male	7	11	Female	2
4	Male	4	12	Female	2
5	Male	7	13	Male	2
6	Male	8	14	Male	1
7	Female	1	15	Male	2
8	Female	2	16	Male	4

**Table 2 ijerph-19-15191-t002:** Correlation coefficients.

Task		Speed	Lon. Acc.	RMSSD	SDNN	LF/HF	HRGR
Phone	Speed	1.000	−0.041	−0.416	−0.365	0.310	0.159
	Lon. Acc.		1.000	−0.300	−0.219	0.174	0.384
	SDNN			1.000	0.659 *	−0.869 *	−0.006
	RMSSD				1.000	−0.575 *	−0.022
	LF/HF					1.000	−0.290
	HRGR						1.000
Slogan	Speed	1.000	0.102	−0.440 *	−0.351	0.103	0.250
	Lon. Acc.		1.000	−0.299	−0.463 *	0.426 *	0.212
	SDNN			1.000	0.812 *	−0.575 *	0.262
	RMSSD				1.000	−0.536*	0.040
	LF/HF					1.000	0.271
	HRGR						1.000
Chatting	Speed	1.000	0.542 *	−0.339 *	−0.025	−0.097	0.538 *
	Lon. Acc.		1.000	0.214	0.416	−0.280	0.350
	SDNN			1.000	0.826 *	−0.537 *	−0.088
	RMSSD				1.000	−0.545 *	0.146
	LF/HF					1.000	−0.394
	HRGR						1.000

*—*p* < 0.05; Lon. Acc.—longitudinal acceleration.

## Data Availability

The data presented in this study are available on request from the corresponding author.

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
