# Peer review of "Effects of Distracting Behaviors on Driving Workload and Driving Performance in a City Scenario"

_ijerph, 2022, doi:10.3390/ijerph192215191_

Round 1

Reviewer 1 Report (Previous Reviewer 2)

The manuscript has largely improved, although conclusions are still poor. Please elaborate at least potential implications for mobility policies, and transferability to other situations, typically what if the study was conducted adapting the tests to senior drivers?

Author Response

Comment: The manuscript has largely improved, although conclusions are still poor. Please elaborate at least potential implications for mobility policies, and transferability to other situations, typically what if the study was conducted adapting the tests to senior drivers?

Response: Thanks for the reviewer’s approval and suggestion. The following paragraph was added in the conclusion section.

Distracted driving occupies visual, cognitive and manual resources that safe driving requires. The mobility policies can propose conversation with car drivers is avoid-able by taking on the mobile phone or in-person. The drivers should be trained to realize the weakness and detrimental habits of driving, and adopt compensatory measures to support safety of the vehicles while distracted. The experimental method including the design of distracting task, test process and data processing is general and transfer-able to other situations, such as senior drivers, motorway and rural scenarios. However, the experimental results in quantification may differ from those presented in this study due to different test scenarios and subjects. Further study will be carried out to comprehensively enhance the conclusions of this paper.

Reviewer 2 Report (Previous Reviewer 3)

I do not have any further comments.

Author Response

Comment: I do not have any further comments.

Response: The authors appreciate the reviewer’s time and approval.

This manuscript is a resubmission of an earlier submission. The following is a list of the peer review reports and author responses from that submission.

Round 1

Reviewer 1 Report

The main contribution of the study is the analysis of heart rate variability and driving performance between distracted and undistracted driving conditions. The manuscript is well organized and comprehensively described. The information provided allows forming an overview on the proposed research. The experimental data analysis appears to be reliable. Please provide some details about future work.

Reviewer 2 Report

The manuscript describes a test on distract driving, which per se is a never-too-explored topic, so thank you for analyzing that. However, in some parts the manuscript is too concise, or edited in a too harsh way. As a result, the reader has no idea of the whole methodological approach, the actual  test development, and which are the implications in terms of road safety, public health or environmental research.

The following is suggested to improve the quality of the paper, which in the present form is not fit for publication. The authors are encouraged to re-submit  the paper, once all of the following have been considered and tackled.

r.80 - " most of the existing distracted driving data have derived from simula- 80 tor-based driving experiments" - please corroborate this statement with references

Introduce a short section where you describe the methodology and what is going to be tested after the introduction. If one does not read the abstract, it is difficult to understand what and how is under measurement and why.

r.99 - the subject group seems to be composed of very young members, or at least the driving experience time suggests that (nobody with at least 10-year experience).....how come? 

section 2.6 - too short...please explain properly each phase in picture 3.....what is the meaning of vi, vj in the figure? please fully clarify. Please also explain what the Savitzky-Golay filtering method is or reference that for those who are not familiar with the method

r.147 - Moreover, not much is told about the driving experience; according to what stated the test seems to take place on a major arterial (6 lanes) in an urban area (which city?). Why was this type of road environment  selected? Why not to select different environments (local, rural)? was it in daytime? peak or off-peak time?. Some more on the test environment would be useful to understand the validity of the test; probably some pictures would help, too.

Moreover, it seems that the authors do not consider the built environment as a distracting factor itself, please elaborate

There is no actual discussion on the results (caveats, potential, replicability, etc.) and the conclusions look like a proxy for a short summary. Please, improve

Reviewer 3 Report

Overall, this paper covers a good topic on distracted driving. I have a few minor comments:

1) I do not feel good when I see the term "cause", in most of the studies on transportation safety, contributing factor is another better term to use;

2) If the subject knows what to happen during the driving, compared to unexpected events, are there any differences? will the results change significantly? especially in the first few seconds;

3) traffic volume is moderate for the experiment duration. How would the authors define moderate? please be specific using some measurements;

4) a little bit of expansion on the Savitzky-Golay filtering method;

5) lines 284-285, reduction of speed indicates greatest driving risk. How can the authors make such a statement? please provide more reasoning.